# Positive Effects of Organic Amendments on Soil Microbes and Their Functionality in Agro-Ecosystems

**DOI:** 10.3390/plants12223790

**Published:** 2023-11-07

**Authors:** Weijia Liu, Zepeng Yang, Qinxin Ye, Zhaohui Peng, Shunxi Zhu, Honglin Chen, Dinghui Liu, Yiding Li, Liangji Deng, Xiangyang Shu, Han Huang

**Affiliations:** 1Institute of Agricultural Bioenvironment and Energy, Chengdu Academy of Agriculture and Forestry Sciences, Chengdu 611130, China; Liuweijia27@163.com (W.L.); yeqinxin123@163.com (Q.Y.);; 2Soil and Fertilizer Research Institute, Sichuan Academy of Agricultural Sciences, Chengdu 610066, China; zepengyang@126.com (Z.Y.); chenhl0107@163.com (H.C.); dinghuiliu@163.com (D.L.); 3College of Resources, Sichuan Agricultural University, Chengdu 611130, China; encili@foxmail.com (Y.L.); auh6@sicau.edu.cn (L.D.); 4Key Lab of Land Resources Evaluation and Monitoring in Southwest, Ministry of Education, Sichuan Normal University, Chengdu 610068, China; 5College of Economics and Management, Xinjiang Agricultural University, Urumqi 830052, China; 18328717493@163.com

**Keywords:** organic amendments, microbes, soil fertility, crop yield, agro-ecosystem, meta-analysis

## Abstract

Soil microbial characteristics are considered to be an index for soil quality evaluation. It is generally believed that organic amendments replacing chemical fertilizers have positive effects on changing microbial activity and community structure. However, their effects on different agro-ecosystems on a global scale and their differences in different environmental conditions and experimental durations are unclear. This study performed a meta-analysis based on 94 studies with 204 observations to evaluate the overall effects and their differences in different experimental conditions and duration. The results indicated that compared to chemical fertilizer, organic amendments significantly increased total microbial biomass, bacterial biomass, fungal biomass, Gram-positive bacterial biomass and Gram-negative bacterial biomass, and had no effect on the ratio of fungi to bacteria and ratio of Gram-positive bacteria to Gram-negative bacteria. Meanwhile, land use type, mean annual precipitation and soil initial pH are essential factors affecting microbial activity response. Organic-amendment-induced shifts in microbial biomass can be predominantly explained by soil C and nutrient availability changes. Additionally, we observed positive relationships between microbial functionality and microbial biomass, suggesting that organic-amendment-induced changes in microbial activities improved soil microbial functionality.

## 1. Introduction

Since the 19th century, the increasing global population and associated food demand have promoted the wide use of chemical fertilizers in agricultural ecosystems. Simultaneously, the annual production of livestock manure and crop residue has also increased rapidly [1,2]. Many studies have found that long-term excessive application of chemical fertilizers, especially nitrogen fertilizers, may exacerbate soil degradation, water pollution, and greenhouse gas emissions [3,4]. In recent decades, organic amendments, such as manure, plant residue and compost, have been considered as an effective strategy to substitute chemical fertilizers because they can improve soil fertility, mitigate greenhouse gas emissions and maintain crop yields [5,6,7,8]. Therefore, the use of organic amendments has attracted worldwide interest, both in research and practical applications.

Soil microorganisms play an important role in ecosystem services and function, including maintaining soil fertility, mitigating soil pollution and regulating soil organic matter decomposition and C, N and P biogeochemical cycles [9,10,11]. Agricultural fertilization has a great impact on microbial activity and community structure, and has received considerable attention. Many studies have shown that compared with no fertilization, the application of organic amendments has a positive effect on microbial activity [12,13]. However, the effects of replacing chemical fertilizers with organic amendments on microbial communities remain unclear. In addition, organic amendments can also affect microbial community structure by changing soil C and N content and accumulation degree, but their impact is still uncertain in comparison with chemical fertilizers [6,14]. More importantly, there are considerable uncertainties regarding the response of microbial activity and community structure to organic amendments under different land use types, types of organic amendments, climate types, soil initial properties and experimental duration. To date, our knowledge of the overall effect of replacing chemical fertilizers with organic amendments on soil microbial activity and community structure is still fragmented. Thus, it is necessary to systematically research this area with independent single studies to compare the effect of organic amendments on soil microbial activity and community structure compared with chemical fertilizers on a global scale, encompassing land use types, organic amendment types, climate conditions, experimental duration and soil properties.

A common notion seems to have developed that microbial community affects microbial function in terrestrial ecosystems [15]. Soil enzyme activities related to soil C, N and P cycling are good proxies of processes driving soil biogeochemical cycling, and are frequently used to estimate the microbial function [16]. However, the relationship between soil microbes and microbial functions under organic amendment is still unclear. These knowledge gaps limit our understanding of the mechanisms by which organic amendments improve microbial function. It is therefore pivotal to study the relationships between the effect of organic amendments on microbial biomass, community and microbial function in order to improve microbial biomass and restore functions in intensive agricultural systems that are routinely fertilized with chemical fertilizer on a global scale.

In this study, we performed a meta-analysis based on 94 studies with 204 observations to evaluate the overall effects of replacing chemical fertilizer with organic amendments on soil microbial activities and community structures in agro-ecosystems. The objectives of this study were to (1) investigate the effects of organic amendments on microbial activity and community structure, (2) identify the potential drivers of microbial activity and community structure responses and (3) reveal the relationship between soil microbes and microbial function related to soil C, N and P cycling.

## 2. Results

### 2.1. Effect of Replacing Chemical Fertilizer with Organic Amendments on Soil Properties, Microbial Function and Yield

Organic amendments significantly increased SOC, TN, AN, AP and MBC on average by 17.22%, 22.09%, 18.15%, 49.09% and 44.86%, respectively, compared with chemical fertilizer. Meanwhile, organic amendments significantly increased soil pH by 25.13%. In addition, organic amendments significantly increased AG, BG, BX, NAG, UREA, BAA, ALP, ACP, DHA and overall microbial function on average by 30.77%, 43.63%, 28.48%, 67.05%, 50.61%, 50.61%, 57.38%, 51.83%, 39.14% and 58.71%, respectively (Figure 1 and Appendix A).

### 2.2. Effect of Replacing Chemical Fertilizer with Organic Amendments on Soil Microbial Activity and Community Structure

Organic amendments significantly increased SOC, TN, AN, AP and MBC on average by 17.22%, 22.09%, 18.15%, 49.09% and 44.86%, respectively, compared with chemical fertilizer. Meanwhile, organic amendments significantly increased soil pH by 25.13%. In addition, organic amendments significantly increased AG, BG, BX, NAG, UREA, BAA, ALP, ACP, DHA and overall microbial function on average by 30.77%, 43.63%, 28.48%, 67.05%, 50.61%, 50.61%, 57.38%, 51.83%, 39.14% and 58.71%, respectively (Figure 1 and Appendix A). In general, organic amendments significantly increased total biomass, bacterial biomass, fungal biomass, actinomycete biomass, Gram-positive bacterial (G+) biomass, Gram−negative bacterial (G−) biomass and arbuscular mycorrhizal fungal (AMF) biomass relative to chemical fertilizer (by 33.40%, 32.71%, 37.93%, 34.04%, 39.49%, 36.48% and 78.75%, respectively; Figure 2 and Figure 3, Appendix A). On average, total biomass, bacterial biomass, fungal biomass, actinomycete biomass, G+ biomass and G− biomass had higher positive responses to the organic amendments in upland soils than in paddy soils. Of all included types of organic amendments, manure had the highest positive effects on total biomass, bacterial biomass, G+ biomass, G− biomass, actinomycete biomass and actinobacterial biomass (38.82%, 42.25%, 41.09%, 57.21%, 51.21%, 32.60% and 135.33%, respectively; Figure 2 and Figure 3, Appendix A), and plant residue had more positive effects on fungal biomass (55.22%) and the fungi-to-bacteria ratio (16.26%). RRs of total biomass and G+ biomass were generally positive regardless of MAP and had the highest values when the MAP was lower than or equal to 500 mm (44.57%, 63.90%, 49.84% and 23.95%, respectively). RRs of bacterial biomass, fungal biomass, actinomycete biomass and AMF biomass were generally positive regardless of MAP and had the highest values when the MAP varied between 500 mm and 1000 mm (42.85%, 69.57%, 59.03% and 139.96%, respectively). Increases in the total biomass, bacterial biomass, fungal biomass and AMF biomass were the largest when the duration of experiments was between 3 years and 10 years (45.50%, 51.42%, 68.40% and 67.60%, respectively), while increases in the actinomycete biomass and G− biomass were the largest in experiments over 30 years in duration (78.91% and 54.71%). Organic amendments also had generally positive effects on total biomass and G− biomass regardless of the soil initial pH, and the highest increases were observed when pH values were higher than 8 (48.08% and 49.05%). Bacterial biomass and G+ biomass responded more positively to the organic amendments at initial pH > 8 (45.97% and 51.35%), and fungal biomass and actinomycete biomass at initial pH 7–8 (89.46% and 109.28%, respectively; Figure 2). In addition, increases in total biomass, bacterial biomass, fungal biomass and actinomycete biomass were more notable when the initial SOC value was lower than or equal to 10 g/kg (35.13%, 38.24%, 53.45% and 51.39%, respectively), and increases in G+ biomass and G− biomass were more notable when the initial SOC value varied between 10 g/kg and 20 g/kg (36.07% and 42.30%, respectively). Moreover, microbial biomass generally showed the highest increment in initial N-depleted soils than in initial N−enriched soils.

Compared with chemical fertilization, organic amendments had no significant effects on the fungi-to-bacteria ratio and the G+-to-G− bacteria ratio (Figure 4). However, we found that plant residue application significantly increased the fungi-to-bacteria ratio (16.26%). Organic amendments significantly increased the G+-to-G− bacteria ratio when MAP was lower than or equal to 500 mm (23.95%), decreased the fungi-to-bacteria ratio in experiments with a duration of more than 30 years (−21.48%), and increased the G+-to-G− bacteria ratio when the duration of experiments was between 3 years and 10 years (12.61%). In addition, the response ratio of the fungi-to-bacteria ratio with organic amendments was higher at initial pH > 8 (22.99%), and the response ratio of the G+-to-G− bacteria ratio was lower than zero when the initial SOC value was higher than 20 g/kg (−19.16%). Response ratio of the fungi-to-bacteria ratio was higher than zero with initial TN ≤ 1 g/kg (10.97%), while it was lower than zero when initial TN was between 1 g/kg and 2 g/kg (−6.19%). Together, these results suggest that organic amendments had more positive effects on soil microbial activity compared with chemical fertilizers and the effects depended on the land use type, MAP and soil pH, but had no effects on microbial community structure.

### 2.3. Key Parameters Influencing the Effect of Replacing Chemical Fertilizer with Organic Amendments on Soil Microbial Biomass

In order to explain the relative importance of soil characteristics, experimental duration and climate factors on soil microbial biomass, we selected the following six indicators for analysis: change in pH, response ratio (RR) of SOC, RR of TN, duration, MAP and MAT. The first four factors of each microbial biomass group could explain the variance in total biomass, bacterial biomass, fungal biomass, the fungi-to-bacteria ratio and the G+-to-G− ratio by 78.10%, 77.82%, 75.49%, 79.74% and 74.56%, respectively (Figure 5). RR of SOC was the predominant factor explaining the variation of total biomass, bacterial biomass and the G+-to-G− bacteria ratio, and MAP and the duration of the experiment were the predominant factors explaining the variation of fungal biomass and the fungi-to-bacteria ratio, respectively, accounting for approximately 22.61%, 21.11%, 24.09%, 22.89% and 24.08%. The correlation analysis indicated that the RR of total biomass (*p* < 0.05), RR of bacterial biomass (*p* < 0.01) and RR of fungal biomass (*p* < 0.05) were significantly increased with the increased RR of SOC (Figure 5).

### 2.4. The Relationships between Soil Microbial Biomass, Microbial Function and Yield

Our correlation analysis showed that the RR of microbial function was significantly and positively correlated with the RR of total biomass (R^2^ = 0.187, *p* < 0.01), RR of bacterial biomass (R^2^ = 0.290, *p* < 0.01) and RR of fungal biomass (R^2^ = 0.215, *p* < 0.01), respectively (Figure 6). Meanwhile, we found that the RR of yield significantly increased with the increasing RR of total biomass (R^2^ = 0.161, *p* < 0.05), but decreased with the increasing RR of the fungi-to-bacteria ratio. Overall, under the conditions of replacing chemical fertilizer with organic amendments, activities of soil microorganisms under organic amendments play an important role in improving microbial function and crop yields.

## 3. Discussion

As a whole, our results found that organic amendments had positive effects on soil microbial biomass compared to chemical fertilizers, which were consistent with the study [2]. Meanwhile, we found that organic amendments had no significant effect on the fungi-to-bacteria ratio and G+-to-G− ratio compared to chemical fertilizer. This result suggested that organic amendments would not consistently select for particular microbial groups in agricultural soils. Importantly, we observed that the responses of microbial activities and community structure to organic amendments varied with land use type, organic amendment types, MAP, experimental duration and soil initial properties. For instance, organic amendments had stronger positive effects on microbial biomass in upland soils than in paddy soils. The explanation was that the anaerobic environment of paddy soils would decrease the rate of microbial degradation and assimilation of organic matter [17,18]. Meanwhile, the higher MAP may limit the promotion effects of organic amendments on nutrient availability and soil C-, N- and P-cycling enzyme activities [6] and consequently reduce microbial C and nutrient use efficiency. As a result, we found that the response of microorganisms to organic amendments was stronger in the areas with lower MAP. Additionally, because of thicker cell walls and capacity to form spores, G+ bacteria have more advantages than G− bacteria under dry conditions, so that organic amendments significantly increased the G+-to-G− bacteria ratio in the areas with lower MAP [19]. We observed that the mean effects of microbial biomass in soils with lower SOC and TN contents were stronger than in soils with higher SOC and TN contents. This suggest that lower initial nutrients have greater microbial C and N limitation and saturation deficit [7], which may result in stronger microbial nutrient demand and higher soil carbon sequestration rate. In addition, numerous previous studies demonstrated that the response of soil microbial community to organic amendments varied with organic material type [20,21,22]. Our results showed that manure application significantly promoted the growth of microbes compared to other fertilizers, which was consistent with a previous meta-analysis study [23]. This could be due to the fact that manure can maintain the stability of soil moisture and temperature environment, while being rich in more readily available C, N and other nutrients required for the growth and activities of microbes [23]. However, we found that the fungi-to-bacteria ratio seemed to respond more strongly to plant residues, which indicates that plant residue was more conducive to the growth and development of fungi than bacteria. This is because these fungi are the main decomposers of these plant residues, and have a dominant position in the early stages of plant residue decomposition compared with other microbes [24]. Our result was consistent with the research results from Aciego Pietri and Brookes [25], who also found that organic amendments caused a higher increase in microbial biomass in alkaline soil. One explanation for this could be that low soil pH may inhibit the activities of most enzymes and whole-cell metabolism [26]. Additionally, the raising of pH caused an increase in availability of organic amendments, so as to improve the absorption and utilization of nutrients by microbes [27].

By grouping the experimental duration, we found that 3, 10 and 30 years were the three most important time periods for the response of a microbial biomass to organic amendments. In general, the microbial biomass responses were quite small during the first three years of organic amendment treatment after which they gradually increased. This could be explained by the fact that the mineralization and nutrient release of organic amendments are relatively slow, and it could take a longer time to produce visible effects [6,28]. However, fungal biomass, G+ biomass and AMF biomass had relatively high responses to organic amendments in the experiments lasting for 3–10 years, and bacterial biomass, G− biomass and actinomycete biomass had relatively high responses to organic amendments in the experiments that lasted more than 30 years. These results indicated that fungal biomass, G+ biomass and AMF biomass had a lower metabolic nutrient demand [29], while bacterial biomass, G− biomass and actinomycete biomass might be more insensitive to organic amendments. In addition, studies have shown that the quantity of labile and refractory substrates of organic matter determined the dominance of fungi and bacteria in the degrader communities, and bacteria dominated microbial communities on high-quality organic matter [29,30]. Thus, long-term application of organic amendments (such as over 30 years) and higher soil initial total nitrogen content could improve soil organic matter quality [31] and then cause fungi to lose their advantages in the competition of substrates, causing a significant decrease in the fungi-to-bacteria ratio. Unlike the fungi-to-bacteria ratio, the response of the G+-to-G− bacteria ratio increased most in the experiments lasting for 3–10 years. The G+-to-G− bacteria ratio may indirectly reflect the relative carbon availability of soil bacterial community carbon in organic soils [32]. Therefore, a large accumulation of soil recalcitrant carbon under organic amendments may preferentially favor the growth of G+ bacteria, thereby increasing the G+-to-G− bacteria ratio.

According to relative importance and linear regression analysis, we found that SOC was the most important factor that affected the responses of microbial biomass to organic amendments. Meanwhile, the effect of soil TN on total microbial biomass was second only to SOC. As a result, SOC and TN play important roles in promoting the growth and development of microorganisms [33]. More importantly, total microbial biomass was positively correlated with microbial function and crop yield. As an important source of soil enzymes, the increase in the biomass of soil microbes enhances the activity of soil enzymes, and then increases the microbial function associated with C-cycling enzymes, N-cycling enzymes, P-cycling enzymes and oxidation enzymes [34,35]. At the same time, the improvement of microbial biomass promotes the decomposition of organic matter and nutrient cycling, and contributes to plant growth [36]. In addition, soil enzymes that reflect microbial function mediate microbes to obtain nutrients from the soil environment, and this process largely depends on the availability of carbon and nitrogen in the soil. These results suggest that organic amendments affected soil microbial activity by regulating soil C and N cycles, and further changed microbial function and crop yields. It is noteworthy that the fungi-to-bacteria ratio negatively affected the crop yield, which might be explained by the effects of an increase in microbial competition due to organic amendment reducing the dependency of a crop on fungi [37].

## 4. Materials and Methods

### 4.1. Data Collection

To evaluate the effect of replacing chemical fertilizer with organic amendments on soil microbial activity and community structure, we collected data from peer-reviewed journal articles published from 2000 to January 2023. We used the ISI Web of Science and China National Knowledge Infrastructure (CNKI) databases and performed a literature search using the terms (organic fertilizer OR organic amendment OR organic input OR organic addition OR manure OR straw OR compost OR waste OR crop residue) AND (microb* OR PLFA OR microbial biomass OR microbial activit*). The following criteria were used to select studies: (1) the experimental duration of the field was at least one year; * (2) the organic amendments and NPK fertilizers were established under the same agricultural management in the field; (3) studies that reported the microbial biomass in both chemical treatments and organic amendment treatments; (4) if the effects of organic amendments on microbial attributes in different crop growing seasons were reported in the study, we only collected data for the last season; (5) we collected the biomass of the microorganisms measured using the phospholipid fatty acids (PLFAs) method; (6) crop yields, soil characteristics (i.e., soil organic carbon (SOC), dissolved organic carbon (DOC), total nitrogen (TN)/phosphorus (TP), available nitrogen (AN)/phosphorus (AP)), soil pH and microbial biomass carbon (MBC) were also collected if they were reported in studies selected in accordance with the above criteria, and if available nitrogen was not reported, we used the sum of ammonium nitrogen and nitrate nitrogen instead; (7) we recoded microbial function associated with C-cycling enzymes, N-cycling enzymes, P-cycling enzymes and oxidation enzymes from the papers, including α-1,4-glucosidase (AG), β-1,4-glucosidase (BG), β-D-cellobiosidase (CBH), β-1,4-xylosidase (BX), invertase (INV), β-1,4-N-Acetyl-glucosaminidase (NAG), leucine aminopeptidase (LAP), urease (UREA), protease (BAA), alkaline phosphatase (ALP), acid phosphatase (ACP), phenol oxidase (PhOx), peroxidase (PEO), catalase (CAT) and dehydrogenase (DHA). The original data of the published articles were extracted from text, tables and graphs. Data in graphs were extracted by GetData Graph Digitizer software. Overall, 204 observations from 94 studies were included in the data collection (Appendix A).

We divided land use types into those that included upland and paddy. Organic amendment types included compost, manure, plant residue, green manure and manure plus plant residue. Soil initial pH, SOC and TN content ranged from 4.07 to 9.20, 1.00 to 29.40 g/kg and 0.02 to 2.24 g/kg, respectively. MAP and experimental duration ranged from 220 to 2600 mm and l to 124 years, respectively. Only 25 studies reported the nitrogen and phosphorus content of organic materials; therefore, we did not analyze the effects of nitrogen and phosphorus content of organic amendments on soil microbial attributes.

### 4.2. Data Analysis

The natural log of response ratio (RR) was used to evaluate the response of target variables to substituting mineral fertilizers with organic amendments. The RR was calculated according to the following equation:(1)$$\ln\mathrm{RR}\;=\;\ln X_{t}-\ln X_{c}$$
where X_t_ and X_c_ denote the mean value of organic amendment and mineral fertilizer treatments, respectively.

Observational variances (ν) were calculated according to the following equation:(2)$$\nu \;=\;\frac{S_{t}^{2}}{N_{t}X_{t}^{2}}+\frac{S_{c}^{2}}{N_{c}X_{c}^{2}}$$
where N_t_ and N_c_ denote the sample size of organic amendment and mineral fertilizer treatments, respectively; S_t_ and S_c_ denote the standard deviations (SD) of organic amendment and mineral fertilizer treatments, respectively. In our datasets, the missing values for each treatment were calculated using the average coefficient of variation (CV) of the datasets which reported the standard deviation or standard error [2].

In this study, we used the random effect model to calculate the overall effect size. For ease of analysis, the ln RR and its corresponding confidence interval were converted back to a percentage change as (e ^ln RR^ − 1) × 100%. The impact of organic amendments on the variable was significantly positive or negative if the 95% confidence intervals (95% CI) of ln RR did not overlap with zero at α = 0.05. Between-group heterogeneity (Q_b_) was examined for a given focused variable to assess the organic amendments effects among the levels of a given variable [23]. Our study used the categorical random effect model to analyze whether microbial biomass showed significantly different responses to organic amendments among different land use types, organic material types, MAP, experimental durations and soil initial pH, SOC and TN. The significant categorical variable (*p* < 0.05) and large Q_b_ values indicated a better ability to predict variation in the overall response ratio compared to other variables in the analysis. The difference in the response ratios between two categories was significant if the 95% CI in the categories were not overlapping. To compare the relative importance of soil properties, climatic (mean annual precipitation (MAP) and mean annual temperature (MAT)) factors and experimental duration on the microbial biomass, the relative influence analysis was calculated. Linear regression analysis and correlation analysis was performed to evaluate the relationship between microbial biomass with yield, soil properties and microbial function under organic amendments. The relative importance analysis and correlation analysis were conducted using the “gbm” and “Performance Analytics” packages, respectively. All statistical analyses were conducted in R program (R version 3.6.3).

## 5. Conclusions

Our comprehensive meta-analysis suggests that organic amendments improved microbial activities, relative to chemical fertilizer, by enhancing SOC and TN in a variety of cropping systems around the world. Meanwhile, positive and significant relationships were found between soil microbial activities, microbial function and crop yield. Notably, soil microbial diversity may influence the performance of organic amendments on microbial-mediated soil functions. However, our study lacked the analysis of microbial diversity. In addition, there are some key parameters, such as soil texture, organic material quality (C/N ratio) and quantity (organic C, N and P input rate) of organic materials, that also greatly affect microbial growth, but these parameters were not included in our meta-analysis due to lack of data. Future research should aim to determine (1) the response of microbial diversity to organic amendments, (2) the relationship between microbial diversity and function and (3) how soil texture, organic material quality and quantity affect the response of the microbial community to organic amendments. Including this information in future studies could help to more comprehensively evaluate the responses of microbial community to organic amendments.

## Figures and Tables

**Figure 1 plants-12-03790-f001:**
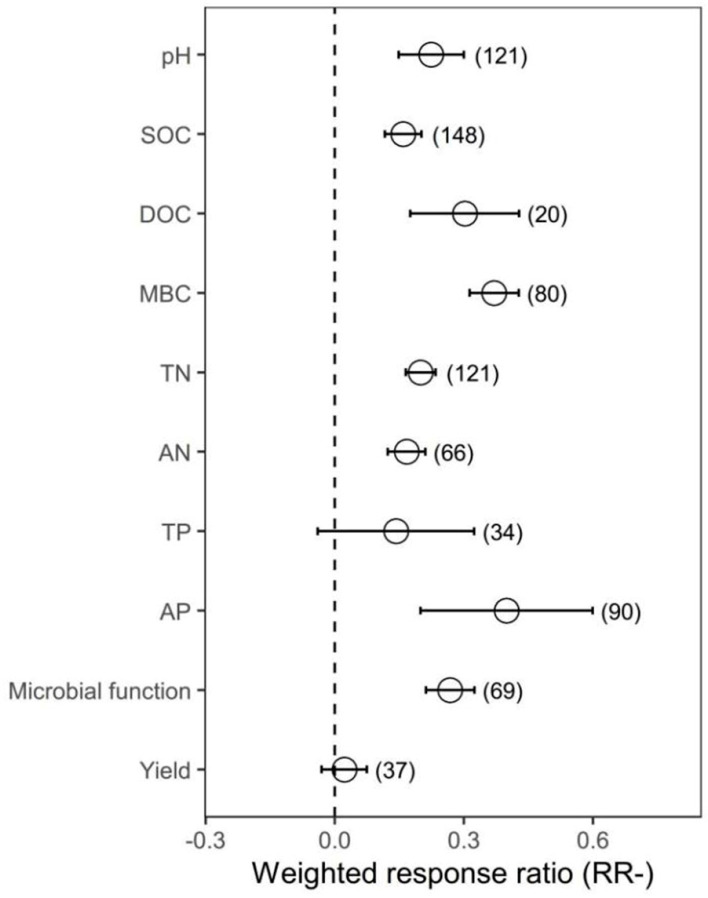
The effect of replacing chemical fertilizer with organic amendments on soil properties, microbial function and yield response ratio (natural logarithm-transformed ratio of organic amendments to chemical treatments, RR). The circles with error bars denote the overall mean response ratio and 95% CI, respectively. The numbers of observations are detailed beside each attribute in parentheses. pH, SOC, DOC, MBC, TN, AN, TP and AP represent soil pH value, organic carbon, dissolved organic carbon, microbial biomass carbon, total nitrogen, available nitrogen, total phosphorus and available phosphorus, respectively.

**Figure 2 plants-12-03790-f002:**
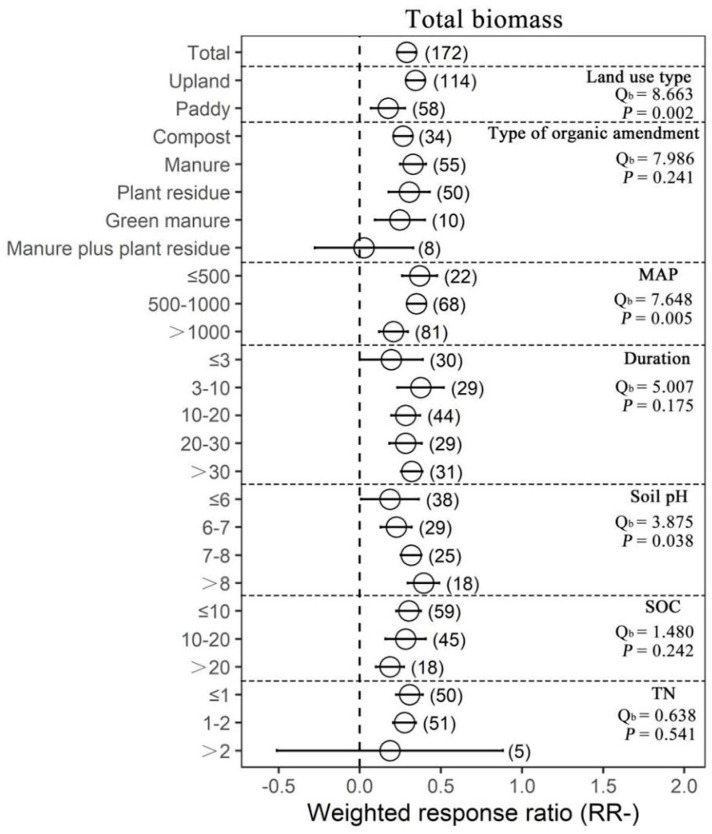
The effect of replacing chemical fertilizers with organic amendments on total biomass response ratio (natural logarithm-transformed ratio of organic amendments to chemical treatments, RR). The circles with error bars denote the overall mean response ratio and 95% CI, respectively. The numbers of observations are detailed beside each attribute in parentheses. The MAP, Duration, SOC and TN denote mean annual precipitation (mm), experimental duration (year), soil organic carbon (g/kg) and total nitrogen (g/kg), respectively. Q_b_ denotes between-group heterogeneity in the same variable and there is significant difference between groups when *p* < 0.05.

**Figure 3 plants-12-03790-f003:**
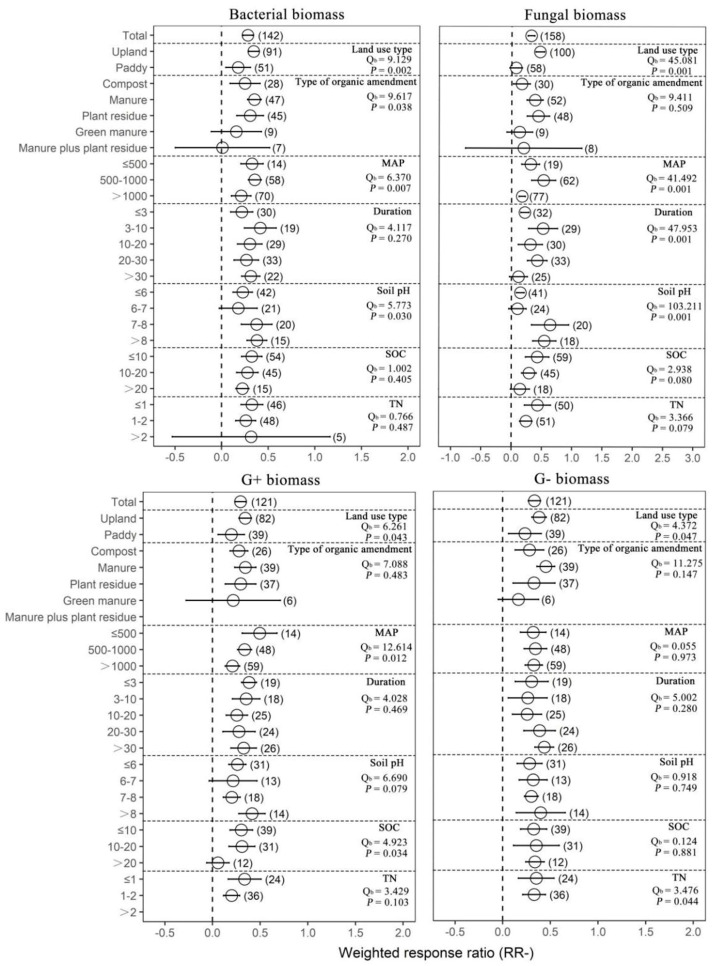
The effect of replacing chemical fertilizers with organic amendments on microbial biomass response ratio (natural logarithm-transformed ratio of organic amendments to chemical treatments, RR). The circles with error bars denote the overall mean response ratio and 95% CI, respectively. The numbers of observations are detailed beside each attribute in parentheses. The MAP, Duration, SOC and TN denote mean annual precipitation (mm), experimental duration (year), soil organic carbon (g/kg) and total nitrogen (g/kg), respectively. Q_b_ denotes between-group variability in the same variable and there is a significant difference between groups when *p* < 0.05.

**Figure 4 plants-12-03790-f004:**
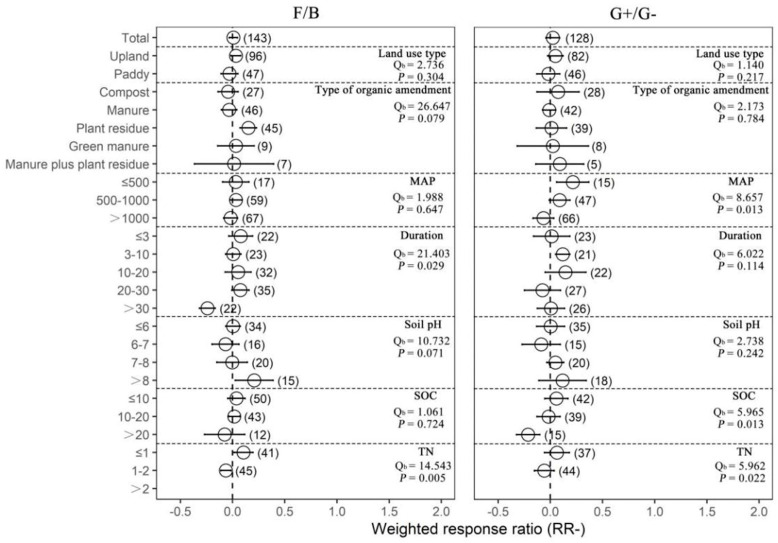
The effect of replacing chemical fertilizer with organic amendments on microbial community structure response ratio (natural logarithm-transformed ratio of organic amendments to chemical treatments, RR). The circles with error bars denote the overall mean response ratio and 95% CI, respectively. The numbers of observations are detailed beside each attribute in parentheses. The MAP, Duration, SOC and TN denote mean annual precipitation (mm), experimental duration (year), soil organic carbon (g/kg) and total nitrogen (g/kg), respectively. Q_b_ denotes between-group variability in the same variable and there is a significant difference between groups when *p* < 0.05.

**Figure 5 plants-12-03790-f005:**
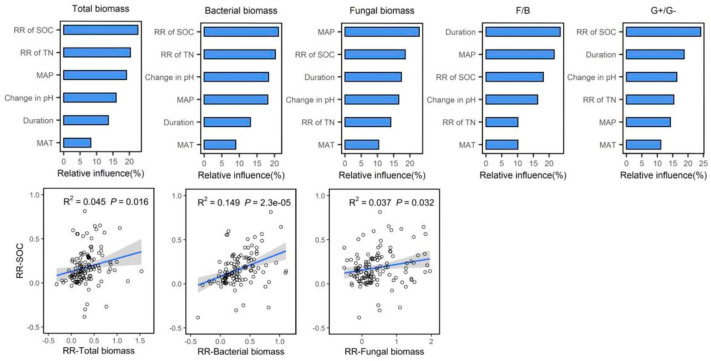
The relative influence (%) of soil properties (pH, SOC and TN), experimental duration and climate (MAP and MAT) on microbial activity and community structure. Correlations between the SOC and microbial activity. The shaded areas show 95% confidence intervals of the fitted regression model.

**Figure 6 plants-12-03790-f006:**
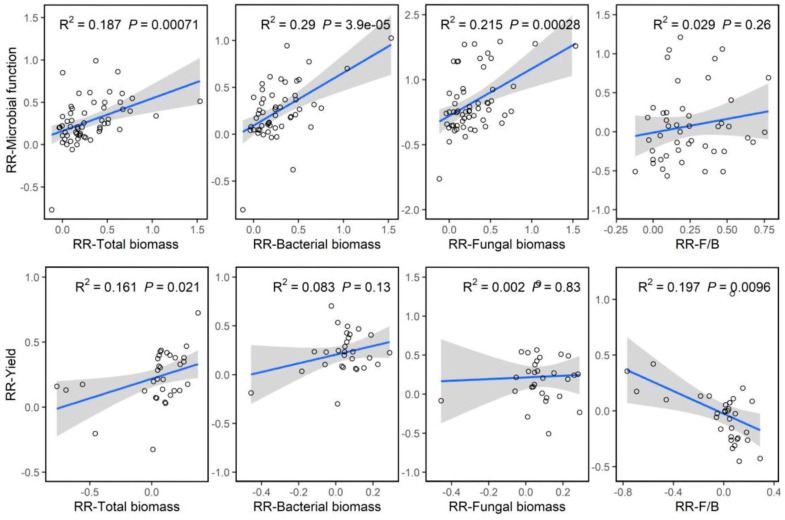
The correlations between the microbial function and microbial activity, community structure and between yield and microbial activity, community structure. The shaded areas show 95% confidence intervals of the fitted regression model.

## Data Availability

Original data may be provided upon request.

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
