# Peer review of "Positive Effects of Organic Amendments on Soil Microbes and Their Functionality in Agro-Ecosystems"

_plants, 2023, doi:10.3390/plants12223790_

Round 1

Reviewer 1 Report

The article is well written

-The downside of the article is that the results are based on a review of publications rather than,  for example, at least 4 years of field studies. The surprising results are :Organic amendments significantly increased SOC, TN, AN, AP and MBC on average 88

- by 17.22%, 22.09%, 18.15%, 49.09% and 44.86%, respectively, compared with chemical fertilizer. ....Meanwhile, organic amendments significantly increased soil pH by 25.13%.

How can such a large increase be explained (line 88-94)? The results and calculations need to be clarified, checked. such a large increase in organic carbon content is highly questionable. A research topic would provide a gap, but it would have to be a field experiment and research. Consider publication from new multi-year studies and compare with existing literature on the subject. The correctness of the results should be checked at this stage. If the data were overstated this should be taken into account in the conclusions.

Author Response

Comments 1: The article is well written.

Response 1: Thank you for reviewing our paper. We appreciate your comments and suggestions to improve the paper for publications.

Comments 2: The downside of the article is that the results are based on a review of publications rather than, for example, at least 4 years of field studies. The surprising results are :Organic amendments significantly increased SOC, TN, AN, AP and MBC on average 88- by 17.22%, 22.09%, 18.15%, 49.09% and 44.86%, respectively, compared with chemical fertilizer. ....Meanwhile, organic amendments significantly increased soil pH by 25.13%.How can such a large increase be explained (line 88-94)? The results and calculations need to be clarified, checked. such a large increase in organic carbon content is highly questionable.

Response 2: Thanks for your comments. We have carefully checked our original data and analysis methods, and we have compared the results of other meta-analysis. Luo et al. (2018) conducted a global meta-analysis and found that organic amendments increased SOC, TN, and MBC on average of 39%, 22%, and 51% than mineral-only fertilization. Du et al. (2020) found that compared to mineral fertilizer, organic amendments increased SOC, TN, AN, and AP on average by 17.7%, 15.5%,16.0, and 66.2%, respectively. All of these studies show that Organic amendments have a positive effect on soil nutrients.

References:

[1] Du, Y., Cui, B., Wang, Z., Sun, J., & Niu, W. (2020). Effects of manure fertilizer on crop yield and soil properties in China: A meta-analysis. Catena, 193, 104617.

[2] Luo, G., Li, L., Friman, V. P., Guo, J., Guo, S., Shen, Q., & Ling, N. (2018). Organic amendments increase crop yields by improving microbe-mediated soil functioning of agroecosystems: A meta-analysis. Soil Biology and Biochemistry, 124, 105-115.

Comments 3: A research topic would provide a gap, but it would have to be a field experiment and research. Consider publication from new multi-year studies and compare with existing literature on the subject. The correctness of the results should be checked at this stage. If the data were overstated this should be taken into account in the conclusions.

Response 3: We appreciate your feedback and understand your concern regarding the absence of empirical experiments in our meta-analysis. We would like to provide an explanation for our choice not to include empirical experiments in this study. The primary aim of our study was to systematically review and synthesize existing literature on effects of organic amendments on soil microbes and their functionality. Given the extensive body of research available on this topic, our goal was to provide a comprehensive overview of the existing evidence rather than generating new experimental data. Meanwhile, meta-analysis is a valuable tool for summarizing and quantifying the collective findings of prior research, which can provide a more robust and nuanced understanding of a topic. While we acknowledge the importance of empirical experiments in scientific research, we believe that our meta-analysis contributes significantly to the field by consolidating and analyzing a substantial body of existing evidence. We believe this approach aligns with the goals and objectives of our study and provides valuable insights for researchers and practitioners in the field. We appreciate your understanding of our approach and the opportunity to clarify our rationale for not conducting empirical experiments in this particular study.

Reviewer 2 Report

The article "Positive effects of replacing chemical fertilizers by organic amendments on soil microbes and their functionality in agro-ecosystems" makes an important analysis on the changes in the soil in the case of replacing chemical fertilizer with organic amendment.The work is interesting, but I have some suggestions: more bibliographic titles would be welcome, and the figures can be explained more clearly.

I have a question related to the 94 studies, where can I find them? are they in the cited bibliography or is there an annex where they are centralized?

Author Response

Thank you very much for taking the time to review this manuscript. Please find the detailed responses below and the corresponding revisions in the re-submitted files.

Comments 1: The article "Positive effects of replacing chemical fertilizers by organic amendments on soil microbes and their functionality in agro-ecosystems" makes an important analysis on the changes in the soil in the case of replacing chemical fertilizer with organic amendment. The work is interesting, but I have some suggestions: more bibliographic titles would be welcome, and the figures can be explained more clearly.

Response 1: Thanks for your comments. Following your suggestion, we have changed the tittle to “Positive effects of organic amendments on soil microbes and their functionality in agro-ecosystems”. Meanwhile, we have explained the figures more clearly.

Comments 2: I have a question related to the 94 studies, where can I find them? are they in the cited bibliography or is there an annex where they are centralized?

Response 2: Thanks for your comments. We have added the dataset of our meta-analysis in supplementary materials. It included references’ name, author’s information and original data.
